# Polymorphism of CLOCK Gene rs3749474 as a Modulator of the Circadian Evening Carbohydrate Intake Impact on Nutritional Status in an Adult Sample

**DOI:** 10.3390/nu12041142

**Published:** 2020-04-19

**Authors:** Marina Camblor Murube, Elena Borregon-Rivilla, Gonzalo Colmenarejo, Elena Aguilar-Aguilar, J. Alfredo Martínez, Ana Ramírez De Molina, Guillermo Reglero, Viviana Loria-Kohen

**Affiliations:** 1Departamento de Farmacología, Facultad de Medicina, Universidad Complutense de Madrid, 28040 Madrid, Spain; mcambl01@ucm.es; 2Nutrition and Clinical Trials Unit, GENYAL Platform IMDEA-Food Institute, CEI UAM + CSIC, 28049 Madrid, Spain; elena.borregon@imdea.org (E.B.-R.); elena.aguilar@imdea.org (E.A.-A.); jalfredo.martinez@imdea.org (J.A.M.); 3Biostatistics and Bioinformatics Unit, IMDEA Food Institute, CEI UAM + CSIC, 28049 Madrid, Spain; gonzalo.colmenarejo@imdea.org; 4Department of Nutrition, Food Science and Physiology, Center for Nutrition Research (CIN), Navarra Institute for Health Research (IdiSNA), 31008 Pamplona, Spain; 5Center of Biomedical Research in Physiopathology of Obesity and Nutrition (CIBEROBN), Institute of Health Carlos III, 28029 Madrid, Spain; 6Molecular Oncology and Nutritional Genomics of Cancer, IMDEA-Food Institute, CEI UAM + CSIC, 28049 Madrid, Spain; ana.ramirez@imdea.org; 7Department of Production and Characterization of Novel Foods, Institute of Food Science Research (CIAL) CEI UAM + CSIC, 28049 Madrid, Spain; guillermo.reglero@imdea.org; 8Production and Development of Foods for Health, IMDEA-Food Institute, CEI UAM + CSIC, 28049 Madrid, Spain

**Keywords:** dietary parameters, carbohydrate intake, obesity, single nucleotide polymorphism, CLOCK gene, rs3749474

## Abstract

The aim of this study was to evaluate the distribution of energy intake and macronutrients consumption throughout the day, and how its effect on nutritional status can be modulated by the presence of the rs3749474 polymorphism of the CLOCK gene in the Cantoblanco Platform for Nutritional Genomics (“GENYAL Platform”). This cross-sectional study was carried out on 898 volunteers between 18 and 69 years old (65.5% women). Anthropometric measurements, social issues and health, dietary, biochemical, genetic, and physical activity data were collected. Subsequently, 21 statistical interaction models were designed to predict the body mass index (BMI) considering seven dietary variables analyzed by three genetic models (adjusted by age, sex, and physical activity). The average BMI was 26.9 ± 4.65 kg/m^2^, 62.14% presented an excess weight (BMI > 25 kg/m^2^). A significant interaction was observed between the presence of the rs3749474 polymorphism and the evening carbohydrate intake (% of the total daily energy intake [%TEI]) (adjusted *p* = 0.046), when predicting the BMI. Participants carrying TT/CT genotype showed a positive association between the evening carbohydrate intake (%TEI) and BMI (β = 0.3379, 95% CI = (0.1689,0.5080)) and (β = 0.1529, 95% CI = (−0.0164,0.3227)), respectively, whereas the wild type allele (CC) showed a negative association (β = −0.0321, 95% CI = (−0.1505,0.0862)). No significant interaction with the remaining model variables was identified. New dietary strategies may be implemented to schedule the circadian distribution of macronutrients according to the genotype. Clinical Trial number: NCT04067921.

## 1. Introduction

Overweight and obesity rates are increasing alarmingly, with excess weight being one of today’s major health issues. Both conditions are characterized by an excessive accumulation of fat in the adipose tissue, hypertrophying and unbalancing the homeostatic processes in which it is involved [1]. This situation increases the risk of prematurely suffering diseases and pathological conditions, such as hypertension, diabetes, cardiovascular events, osteopathy, and cancer; being one of the main causes of population’s morbidity and mortality [2,3,4].

As for the prevalence worldwide, it is estimated that approximately 30% suffer from obesity and 35% show a body mass index (BMI) in overweight ranges [3]. In Spain, according to the National Statistics Institute, up to 62.5% of men and 46.7% of women have an excess body weight (BMI >25 kg/m^2^) [5].

Obesity treatment strategies traditionally focus on weight loss by combining dietary and physical activity recommendations as well as pharmacological and surgical therapeutic approaches [6]. Although many dietary alternatives are available for weight control, such as low-carbohydrate diet and low-fat diet, it is not known which is the most effective [7]. In recent years, a more detailed study of dietary strategy has been promoted, not only based on the amount of kilocalories or macronutrients provided, but also on their circadian distribution throughout the day. Among the various studies, some show a benefit of carbohydrate consumption mostly displaced towards late-night hours during dinner [8,9,10]. Others, however, support that the intake of this macronutrient in breakfast or lunch has a metabolic benefit as well as on satiety [11,12]. Indeed, the effectiveness of the distribution of calories and macronutrients over the day is controversial. While some of these discrepancies may be explained by methodological differences in the studies, they may also be due to genetic modulation of the response. In this context, single nucleotide polymorphisms (SNPs) constitute the majority of human genomic variation among individuals. These variations in DNA sequence can affect the response of individuals to drugs, viruses, bacteria, and also to diet [13,14].

Recent research has linked the presence of certain polymorphisms in genes such as *CLOCK* (Circadian Locomotor Output Cycles Kaput) to different responses to caloric and macronutrient intake. Regarding *CLOCK* rs3749474, it has been shown that both TT+CT genotypes tended to have a better weight loss after a treatment involving dietary fat restriction than those homozygous for the wild type allele (CC) [15]. However, the evidence about the mode this SNP could influence the response according to the distribution of different macronutrients throughout the day is unclear [16].

On this basis, the aim of this study was to evaluate the distribution of energy intake and macronutrients throughout the day, and how its effect on nutritional status can be modulated by the presence of the *CLOCK* SNP rs3749474 in the sample of the Cantoblanco Platform for Nutritional Genomics (GENYAL Platform). The final objective was to figure out some of the controversies found in the dietary recommendations and describe new alternatives for a personalized nutritional treatment for obesity.

## 2. Materials and Methods

### 2.1. Study Sample and Design 

This observational and cross-sectional study was carried out on volunteers belonging to the Cantoblanco Platform for Nutritional Genomics (GENYAL Platform) recruited during the period 2012–2017 in Spain. This platform is a tool at the IMDEA-Food Institute for the study of genome-nutrient interactions on a large scale.

Recruitment was developed through media and included volunteers (men and women), aged between 18 and 69 years of age, who did not suffer from any serious diseases, were pregnant or lactating, who willingly signed the consent participation form.

The study was approved by the Research Ethics Committee of the Autonomous University of Madrid (CEI 27-666) and was adapted to the ethical bases proposed by the Declaration of Helsinki with regard to scientific research studies and valid laws. 

### 2.2. Personal, Social, and Health Data 

The information that was collected included date of birth, sex, ethnicity, marital status, level of education, and employment status, as well as health information, such as smoking, alcohol consumption, minor illnesses, medication, and family history of disease. 

### 2.3. Anthropometric Parameters

Anthropometric measurements were performed under standardized procedures. Height (cm) was assessed to the nearest 0.1 cm using a stadiometer (Leicester Biológica Tecnología Médica SL, Barcelona, Spain). Weight (kg), fat mass (%), and muscle mass (%) were evaluated using bioelectrical impedance analysis (Body Composition Monitor BF511-OMRON HEALTHCARE, LT, Kyoto, Japan). Based on these data, the BMI was calculated according to the Quetelet Index ($\frac{Weight\;\left(kg\right)}{Height\;{\left(m\right)}^{2}}$). The World Health Organization’s criteria (WHO) [17] was used to classify the subjects and were regrouped as normal weight (NW) when BMI < 25 kg/m^2^ and as excess weight (EW) when BMI ≥ 25 kg/m^2^. Waist and hip circumference were measured with a flexible Dry 201 metal tape, with measuring range 0–150 cm and 1 mm of precision (Quirumed, Valencia, Spain). 

### 2.4. Dietary Parameters

Regarding the study of the dietary intake, volunteers completed a validated three-day food record prior to the visit (two weekdays and another weekend day) in which they wrote all the food and beverages consumed, as well as the exact or estimated weight in measurements at home (for which they were instructed by trained nutritionists). Subsequently, the extracted data were tabulated and analyzed using the DIAL nutritional software (version 3.7.1.0 (February 2019)-Alce Ingeniería, Madrid, Spain) as described by the supplier [18]. 

Based on the Spanish food pattern, three main meals (breakfast, lunch, and dinner) and three in-between meals eating occasions (morning, evening, and after dinner snack) were considered. The intake at mid-morning, mid-afternoon, and late-evening occasions in this population is usually a light and easily prepared meal, such as a sugar-sweetened infusion (tea, coffee…), a piece of fruit, a dairy product, cookies and similar products, or reasonable combinations of them. 

Multiple dietary variables were considered in the exploratory analyses. The Total Daily Energy Intake (TEI), the Meal Energy Intake (MEI), and the percentages of macronutrients (carbohydrates (CH), proteins (Pr), and lipids (Lip)) of TEI (CH (%TEI), Pr (%TEI), Lip (%TEI), as well as of each meal (CH (%MEI), Pr (%MEI), Lip (%MEI)) were automatically calculated by the DIAL software. 

In addition, to account the timing distribution of energy and the macronutrients of the TEI, the following variables were created:

Meal Energy Intake (MEI) as a percentage of TEI:$$\mathrm{MEI}\;\left(\%\mathrm{TEI}\right)=\frac{\mathrm{MEI}\;\left(kJ\right)}{\mathrm{TEI}\;\left(\frac{kJ}{day}\right)}\;\times \;100.$$

Macronutrients (CH, Pr, Lip) of each MEI over TEI:$$Meal\;\mathrm{CH}\;\left(\%TEI\right)=\frac{\mathrm{CH}\;of\;\mathrm{MEI}\;\left(kJ\right)}{\mathrm{TEI}\;\left(\frac{kJ}{day}\right)}\;\times \;100\;;\;Meal\;\mathrm{Pr}\;\left(\%\mathrm{TEI}\right)=\frac{\mathrm{Pr}\;of\;\mathrm{MEI}\;\left(kJ\right)}{\mathrm{TEI}\;\left(\frac{kJ}{day}\right)}\;\times \;100;\;\;Meal\;\mathrm{Lip}\;\left(\%\mathrm{TEI}\right)=\frac{\mathrm{Lip}\;of\;\mathrm{MEI}\;\left(kJ\right)}{\mathrm{TEI}\;\left(\frac{kJ}{day}\right)}\;\times \;100.$$

In order to investigate if a higher intake of energy and macronutrient at the beginning or the end of the day may influence the nutritional status, the meals were regrouped into two sub-groups according to the time-of-day: morning intakes (MI) (including breakfast and morning snack) and evening intakes (EI) (including evening snack, dinner, and after dinner snack). 

An initial inspection of the data showed a discordance between the information provided by some subjects in the dietary questionnaire and their anthropometric parameters. Thus, some with high BMI, body fat, and high levels of blood pressure, reported lower intake in terms of total calories, as well as healthier and more balanced nutritional profiles. This finding suggested that some of these individuals could be underreporting their dietary data. To overcome this limitation, the method proposed by the European Food Safety Authority (EFSA) [19] and Goldberg and Black [20,21] was followed. This method assumes that, if weight remains stable, energy intake should be equal to energy expenditure. According to this procedure, the volunteers were classified as underreporting, overreporting and correct reporting, creating a final classification of two groups, namely misreporters and plausible reporters. The exploratory analysis of the dietary and intake data was performed with both the total sample and the plausible reporters. However, the statistical models (see below) were only derived for the plausible reporters.

### 2.5. Physical Activity Parameters

The short version of the International Physical Activity Questionnaire (IPAQ) validated in the Spanish population [22] was used. According to the results, the pattern of activity was classified into three categories: low, moderate, and high, according to the authors proposal [22].

### 2.6. Biochemical Parameters and Genotyping

A blood sample was collected by venipuncture in the middle cubital vein of the forearm early in the morning, with the subject fasting for at least the previous 12 h. 

The CQS Laboratory, following standardized procedures, carried out the analysis of the biochemical determinations. The lipid profile [(triglycerides, total cholesterol, low-density lipoprotein cholesterol (LDL-cholesterol), and high-density lipoprotein cholesterol (HDL-cholesterol)] were determined by enzymatic spectrophotometry. The glycemic profile [glycemia (c. hexokinase mass) and insulin (immunoassay)] and the Homeostasis Model Assessment (HOMA) was calculated regarding the following formula:$$\mathrm{HOMA}=\frac{Glucose\left(\frac{mg}{dL}\right)\;\times \;Insulin\;\left(\frac{\mu U}{mL}\right)}{405}.$$

The presence of the rs3749474 *CLOCK* polymorphism was determined at the IMDEA-Food Nutritional Genomics Laboratory from these blood samples. This SNP was selected based on its recognized involvement in different parts of the pathogenic processes of obesity and its related phenotypes, and because it has been associated with different genotype response to weight loss in a previous study [15]. Preserved at −80 °C. DNA extraction was performed using the QIAamp DNA Mini Kit, QIAGEN, following the protocol of the commercial company, in which 300 µL of sample was used to obtain 100 µL of DNA, establishing the optimal cut points for DNA concentration at 50 ng/µL and 1.7 of quality (absorbance ratio A260/A280 and A260/A230). Once the DNA had been extracted, genotyping was performed using TaqMan^®®^ SNPs probes with the QuantStudio™ 12K Flex Real-Time PCR System and the AccuFill™ system. Data analysis was performed using TaqMan Genotyper Software v1.3. A quality value of each genotyping of more than 90% was used.

### 2.7. Statistical Analysis

Categorical data were presented as percentages and absolute frequencies, while quantitative data were expressed as mean ± standard deviation (SD). Linear regression models adjusted by sex and age were used to assess associations between anthropometric, dietary, and physical activity variables. In order to test the interaction between the rs3749474 SNP and nutritional variables in the prediction of BMI, the following seven evening variables were considered: Evening Meal Energy, Carbohydrates, Proteins, and Lipids Intake of the Total Daily Energy Intake [Eve MEI (%TEI), Eve CH (%TEI), Eve Pr (%TEI) and Eve Lip (%TEI)], and Evening Carbohydrates, Proteins, and Lipids Intake of the Meal Energy Intake [Eve CH (%MEI), Eve Pr (%MEI), and Eve Lip (%MEI)]. In addition, three genetic models (additive (ADD), co-dominant (COD), and dominant (DOM)) were considered, giving a total of 21 models, all of them adjusted by sex, age, and physical activity (expressed in metabolic equivalents (METs)). It was finally determined that the ADD model (in which the risk conferred by an allele is increased 2-fold for homozygotes) was the best fit for the data. *p* Values were corrected with Bonferroni’s method. All statistical tests were considered as bilateral with a significance level of 0.05. Estimated parameters (Betas) were obtained with 95% confidence intervals. Statistical analyses were performed using R version 3.4 (projects) (www.r-project.org).

Sample size calculations were performed with G*Power 3.1.9.2. for a multiple regression model of 6 predictor variables. For an effect size of 0.01 for adding a new single variable, with a power of 0.8 and a significance level of 0.05, a sample size of 787 was obtained. Assuming a drop-out of 15%, this gave a sample size of 905 individuals. 

## 3. Results

The sample was composed of 898 subjects, of which 65.5% were women and 34.5% were men. The mean age of the sample was 41 ± 12 years and the BMI 26.9 ± 4.65 kg/m^2^. About 62.14% of the volunteers presented excess weight (EW, BMI >25 kg/m^2^). Anthropometric, biochemical, physical activity, as well as dietary data of the participants according to their nutritional status is provided in the Appendix A section. A total of 799 subjects had dietary data available and 697 were considered according to Goldberg’s method. Finally, 84% were classified as plausible reporters (*n* = 585) and 16% as misreporters (*n* = 112). 

When the BMI was regressed on the Total Daily Energy Intake (TEI), a positive and significant association was observed (β = 0.00132, 95% CI = (0.000505,0.00213); *p* = 0.038). On the other hand, the percentages of macronutrient (CH, Pr, and Lip) as a percentage of TEI showed no significant associations with the BMI of the total participants (*p* > 0.05).

In terms of energy and macronutrient intake per meal (breakfast, morning snack, lunch, evening snack, dinner, and after dinner snack, morning and evening) no significant association with BMI was identified. Nevertheless, morning carbohydrate intake (%TEI) was associated with different markers of the glycemic profile: glycemia (β = −0.419, 95% CI = (−0.628, −0.21); *p* = 0.001), HOMA (β = −0.0443, 95% CI = (−0.0707, −0.0178); *p* = 0.014) and insulin (β = −0.181, 95% CI = (−0.288,−0.0743); *p* = 0.012) and lipid: LDL (β = −1.09, 95% CI = (−1.69,−0.488); *p* = 0.005) and TG (β = −0.813, 95% CI = (−1.32,−0.307); *p* = 0.022). 

Table 1 describes the main anthropometric and biochemical data of the sample split by genotypes of the rs3749474 polymorphism, as well as the *p* value of a linear model of the SNP as predictor adjusted by sex and age. Table 2 includes dietary data for the total and the time-of-day intakes (breakfast, morning snack, lunch, evening snack, dinner, and after dinner snack, morning and evening intake) in plausible reporters by genotypes. The *p* value of a linear model of the SNP as predictor adjusted by sex and age is also included. 

In order to better understand the genetic dependence of the effect of the distribution of meals in the nutritional status, 21 different lineal models were developed to predict BMI. Each of them had an interaction term between the SNP and one of the following 7 evening variables: Evening Meal Energy, Carbohydrates, Proteins, and Lipids Intake, of the Total Daily Energy Intake [Eve MEI (%TEI), Eve CH (%TEI), Eve Pr (%TEI), and Eve Lip (%TEI)], and Evening Carbohydrates, Proteins, and Lipids Intake of the Meal Energy Intake [Eve CH (%MEI), Eve Pr (%MEI) and Eve Lip (%MEI)]. In addition, for each evening variable, three different genetic models for the SNP were tested: additive (ADD), co-dominant (COD), and dominant (DOM)). All the models were adjusted by sex, age, and physical activity (expressed in metabolic equivalents (METs). 

After multiple-test correction by Bonferroni method, a significant interaction was observed between the presence of the rs3749474 polymorphism in the additive model and the evening carbohydrate intake (*p* = 0.046) when predicting the BMI (Figure 1). The heterozygous subjects (CT) showed that for every 1% increase in carbohydrate intake of the total energy intake during evening hours, the BMI increased by 0.1529 kg/m^2^ (β = 0.1529, 95% CI = −0.0164,0.3227)) and the homozygotes for the risk allele (TT) showed an even greater variation, increasing the BMI by 0.3379 kg/m^2^ for each 1% increase in carbohydrate consumption during this time (β = 0.3379, 95% CI = (0.1689,0.5080)). However, the common homozygous (CC) did not experience statistic increases in the BMI, even showing a slight decrease in it (−0.0321 kg/m^2^), (β = −0.0321, 95% CI = (−0.1505,0.0862)). Moreover, no significant interaction with the remaining model variables was identified. 

## 4. Discussion

The relationship between the food intake at different times of the day and the nutritional status in a group of volunteers belonging to the GENYAL Platform sample was studied in the present research. A significant interaction between evening intake of carbohydrates of the volunteers and the rs3747494 *CLOCK* polymorphism when predicting BMI was identified, suggesting new alternatives of personalized nutritional treatment for obesity.

In this research, excess weight people had defective lipid and glycemic profiles in relation to normal weight people. Furthermore, this group showed lower levels of physical activity (these results are provided in the Appendix A section). These features of excess weight people have been well described in the scientific literature [4,23,24] and have been associated with an increase of concomitant diseases, such as type II diabetes mellitus, cancer, etc. [2,11,25], and thus, emphasize the importance of managing weight control in order to avoid secondary clinical complications.

The total daily energy intake has been postulated as a regulator of the nutritional status, especially if it is not combined with physical exercise according to the energy intake [26,27]. This positive association between the energy intake and BMI was observed, but only after considering the plausible reporters. This highlights the importance of verifying the reliability of dietary intake data using the proposed methods by Goldberg and Black in order to minimize the bias generated by participants reporting [28].

Furthermore, we observed that morning carbohydrate intake was associated with better lipid and glycemic profiles. This finding corroborates previous studies [11,12] that support better nutritional and health status when a higher proportion of carbohydrates is made in the first half of the day. These studies suggest that a higher intake in the morning hours translates into a lower overall intake compared to main meals eaten later. It has also been suggested that this behavior could be influenced by the levels of ghrelin, a hormone that is involved in controlling appetite, since elevated levels of ghrelin were found during the early part of the night, decreasing in the morning before awakening [29]. In addition, according to these studies, insulin sensitivity and glucose tolerance decrease progressively throughout the day [11] with a negative effect on weight management. 

Conversely, other authors reported better results in weight control when the highest intake of carbohydrates is made in late hours [8,9,10]; however, this was not observed in this study. The authors who support late high carbohydrates intake postulate a benefit in maintaining the levels of leptin with a consequent effect on satiety. Moreover, they argue that the consumption of nocturnal carbohydrates is related to higher levels of adiponectin, which results in better insulin sensitivity and a better inflammatory profile [9]. 

Given the controversy about the association between carbohydrate consumption during the evening and the nutritional status, it was planned to investigate this association by considering the SNP rs3749474 of the *CLOCK* gene as a modulator. A significant interaction was observed between the intake of carbohydrates displaced at evening hours (evening snack, dinner, and after dinner snack) and the SNP genotype when predicting the BMI. The higher intake of carbohydrates in the evening was associated with a higher BMI in volunteers with one or more risk alleles, a situation not observed in wild type. The *CLOCK* gene is a regulator of circadian rhythms that modulates the expression of PPAR (Peroxisome Proliferator-Activated Receptor), which corresponds to a family of transcription factors involved in cellular lipid metabolism (lipolysis and lipogenesis), whose activity is regulated following circadian cycles [30,31]. The presence of the polymorphism of the *CLOCK* rs3749474 gene implies the exchange of a cytosine (C) for a thymine (T) on the 3’-non-coding region of the gene. This situation affects the folding and stability of its mRNA, compromising its functionality and the factors it regulates, such as PPAR [15,32] so that the processes of lipogenesis and lipolysis are altered and lose effectiveness, influencing fat deposition. 

Several studies have investigated the relationship between fat-rich diets and the deregulation of circadian rhythms [31,33]. It has been observed that the presence of alterations in the *CLOCK* gene implies a modification in the normal patterns of secretion of neuropeptides involved in the appetite/satiety pathways. Consequently, the presence of the polymorphism rs3749474 has been associated with higher energy intake [34]. However, the influence that this polymorphism can have in the context of a higher consumption of carbohydrates in late hours has been less studied, as is shown by the interaction found in this work.

The metabolism of carbohydrates and lipids is strongly linked in a context of the overabundance of glucose (e.g., high-carbohydrate diet), excess of glucose is metabolized to lipids for storage [35]. In this way, a high-calorie diet based on carbohydrates can lead to an increase in total body fat. In addition, the presence of polymorphism in the *CLOCK* rs3749474 gene has been associated with higher levels of orexigenic hormones, such as ghrelin, and lower levels of anorexigenic hormones, such as leptin, leading to physiological disorders in circadian patterns [33]. Under normal conditions, which one release is increased during the late hours. Nevertheless, this polymorphism influences maintaining low levels of leptin during this period inducing greater appetite. Deregulation of insulin secretion has also been observed in previous studies when *CLOCK* is altered [31]. 

We hypothesized then that all these altered mechanisms described in previous studies could contribute to a poor energy management in the presence of the SNP and explain in part the negative effect on the nutritional status. The evening energy and carbohydrate intake showed no difference due to genotype. These data suggest that the effect on BMI may be more related to the metabolic disorders associated with the presence of the SNP than to a greater appetite.

Given that the volunteers carrying TT alleles presented a worse nutritional status, we suggest that controlling the amount of carbohydrates consumed in the evening could be an approach for personalizing nutrition strategies in this group. Individuals with CT and TT genotypes would be recommended carbohydrate intakes mostly at morning hours in order to avoid the negative effect on the nutritional state. This finding highlights the interest of studies on gene-diet interaction, to contribute to dietary interventions aimed at the normalization of nutritional status. 

This work is an observational study based on self-reported dietary surveys; therefore, this data is subject to under/overreporting. In order to overcome this limitation, people whose weight stability matched their reported energy intake were included and described as ‘plausible reporters’.

More research is needed to confirm these findings, through prospective studies that include different interventions after genotype and compare different dietary approaches to the carbohydrate distribution throughout the day, taking into account the presence of this SNP.

## 5. Conclusions

Evening carbohydrate intake in the presence of the rs3747494 polymorphism in homozygosis or heterozygosis is associated with a higher BMI. In accordance with this evidence, the response to late-time carbohydrate consumption should consider a sufficient knowledge of genetic factors. It seems reasonable to hypothesize that the controversy regarding the correct pattern of intake could be directly related to the genotype studied, since previous studies have not considered the influence of SNPs. As a new nutritional intervention tool, dietary recommendations for SNP CLOCK carriers should be aimed at distributing carbohydrates mainly at morning hours for a personalized circadian nutrition.

## Figures and Tables

**Figure 1 nutrients-12-01142-f001:**
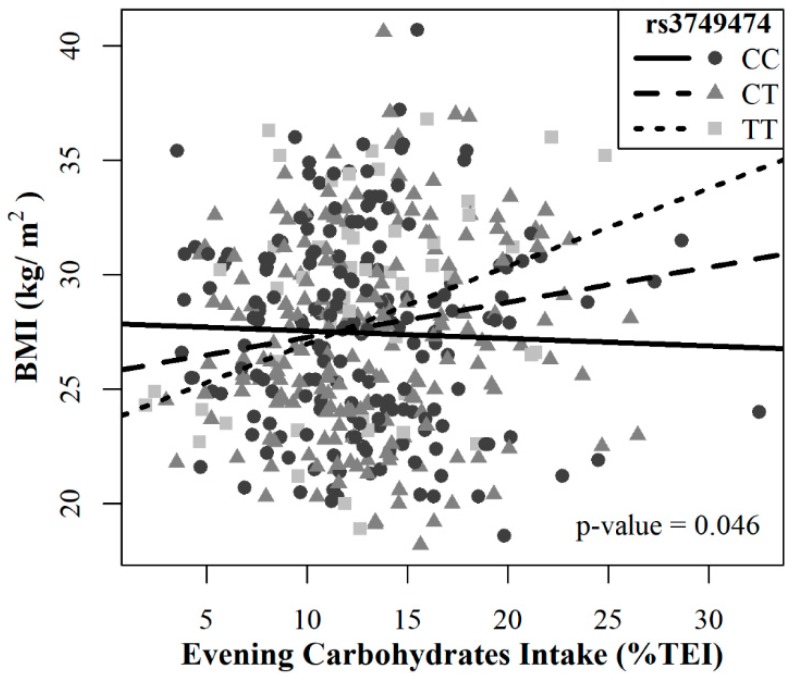
Association between the evening carbohydrates intake as a percentage of Total Daily Energy Intake (TEI) and the body mass index (BMI) according to the presence of the rs3749474. Participants carrying TT and CT genotype showed a positive association between the evening carbohydrate intake (%TEI) and the BMI (β = 0.3379, 95% CI = (0.1689,0.5080)) and (β = 0.1529, 95% CI = (−0.0164,0.3227)) respectively, whereas the common homozygous (CC) showed a negative association (β = −0.0321, 95% CI = (−0.1505,0.0862)) (adjusted *p* = 0.046).

**Table 1 nutrients-12-01142-t001:** Anthropometric and biochemical data by genotypes of the rs3749474.

	Genotype			
Variable	CC (*n* = 271)	CT (*n* = 283)	TT (*n* = 70)	*p* Value
BMI (kg/m^2^)	28.00 (27.44,28.56) ^1^	27.41 (26.86,27.96)	28.84 (27.76,29.92)	0.048 *
Total Fat Mass (%)	33.72 (32.81,34.62)	32.93 (32.04,33.82)	34.87 (33.13,36.62)	0.115
WC (cm)	92.59 (91.02,94.15)	91.29 (89.75,92.84)	96.21 (93.18,99.25)	0.016 *
SBP (mm Hg)	124.70 (123.01,126.39)	124.78 (123.11,126.45)	125.62 (122.34,128.91)	0.880
DBP (mm Hg)	77.40 (76.25,78.55)	77.55 (76.41,78.69)	78.84 (76.6,81.07)	0.516
HDL (mg/dL)	51.73 (50.21,53.26)	52.09 (50.59,53.59)	53.37 (50.41,56.33)	0.619
LDL (mg/dL)	127.76 (124.02,131.49)	130.35 (126.66,134.04)	131.09 (123.86,138.32)	0.531
TG (mg/dL)	103.46 (95.97,110.95)	104.23 (96.88,111.58)	115.31 (100.84,129.79)	0.332
LDL/HDL	2.60 (2.5,2.71)	2.64 (2.54,2.74)	2.62 (2.42,2.82)	0.877
TC/HDL	4.08 (3.95,4.21)	4.13 (4.01,4.26)	4.11 (3.87,4.36)	0.818
Log TG/HDL	0.25 (0.22,0.29)	0.25 (0.22,0.28)	0.30 (0.24,0.36)	0.370
Glucose (mg/dL)	87.19 (85.72,88.67)	85.63 (84.17,87.1)	88.22 (85.53,90.92)	0.144
HOMA	2.03 (1.85,2.21)	1.79 (1.61,1.98)	2.08 (1.74,2.43)	0.122
Insulin (µUI/mL)	9.07 (8.38,9.76)	8.34 (7.65,9.03)	9.53 (8.23,10.82)	0.162

Hardy–Weingber equilibrium *p* = 0.8157. ^1^ Mean (95% CI). * *p* < 0.05 is considered as statistically significant. CC: Common Homozygous; CT: Heterozygous; DBP: Diastolic Blood Pressure; HOMA: Homeostatic Model Assessment; SBP: Systolic Blood Pressure; TC: Total Cholesterol; TT: Variant Homozygous; WC: Waist Circumference.

**Table 2 nutrients-12-01142-t002:** Dietary data for the total and the time-of-day intakes by genotype of the rs3749474 (plausible reporters).

	Genotype			
	CC (*n* = 182)	CT (*n* = 196)	TT (*n* = 49)	
Dietary Data	X ± SD	X ± SD	X ± SD	*p* Value
**Total dietary data**				
TEI (kJ/day)	9450 ± 1948	9215 ± 2024	9729 ± 2126	0.159
CH (%TEI)	37.97 ± 6.51	38.44 ± 6.03	36.91 ± 6.67	0.305
Pr (%TEI)	17.59 ± 2.91	17.57 ± 2.81	17.09 ± 2.86	0.717
Lip (%TEI)	40.38 ± 6.00	39.64 ± 5.92	41.05 ± 6.26	0.250
**Dietary data: Breakfast**				
B MEI (%TEI)	16.71 ± 6.54	16.66 ± 6.12	16.60 ± 8.09	0.994
B CH (%TEI)	8.89 ± 3.77	8.86 ± 3.44	8.23 ± 4.55	0.392
B Pr (%TEI)	2.27 ± 1.14	2.27 ± 1.01	2.33 ± 1.25	0.882
B Lip (%TEI)	5.12 ± 2.95	5.1 ± 3.02	5.61 ± 3.08	0.395
**Dietary data: Morning snack**				
MS MEI (%TEI)	6.38 ± 5.69	5.94 ± 4.67	7.44 ± 6.07	0.437
MS CH (%TEI)	3.09 ± 2.90	3.01 ± 2.42	3.36 ± 2.70	0.634
MS Pr (%TEI)	0.85 ± 0.97	0.79 ± 0.82	1.09 ± 1.11	0.478
MS Lip (%TEI)	2.08 ± 2.41	1.76 ± 1.98	2.46 ± 2.40	0.320
**Dietary data: Lunch**				
L MEI (%TEI)	39.77 ± 8.61	40.08 ± 8.88	39.34 ± 9.32	0.745
L CH (%TEI)	13.31 ± 4.49	13.50 ± 4.55	12.62 ± 4.36	0.427
L Pr (%TEI)	7.91 ± 2.24	7.92 ± 2.17	7.37 ± 2.26	0.186
L Lip (%TEI)	16.96 ± 5.62	16.93 ± 5.79	17.32 ± 5.28	0.818
**Dietary data: Evening snack**				
ES MEI (%TEI)	7.06 ± 6.10	6.60 ± 5.59	6.70 ± 5.07	0.798
ES CH (%TEI)	3.25 ± 2.90	3.17 ± 2.78	3.20 ± 2.37	0.821
ES Pr (%TEI)	0.95 ± 0.98	0.90 ± 0.96	0.87 ± 0.87	0.747
ES Lip (%TEI)	2.58 ± 2.77	2.23 ± 2.51	2.28 ± 2.35	0.512
**Dietary Data: Dinner**				
D MEI (%TEI)	29.25 ± 8.55	29.59 ± 9.86	29.14 ± 8.39	0.918
D CH (%TEI)	9.14 ± 3.85	9.43 ± 4.11	9.15 ± 3.86	0.888
D Pr (%TEI)	5.45 ± 1.89	5.55 ± 2.00	5.34 ± 2.25	0.660
D Lip (%TEI)	13.31 ± 4.99	13.21 ± 5.48	13.12 ± 4.59	0.967
**Dietary data: After dinner snack**				
ADS MEI (%TEI)	0.85 ± 2.66	1.14 ± 3.26	0.78 ± 2.74	0.407
ADS CH (%TEI)	0.29 ± 0.87	0.48 ± 1.57	0.35 ± 1.29	0.412
ADS Pr (%TEI)	0.15 ± 0.73	0.13 ± 0.42	0.09 ± 0.33	0.462
ADS Lip (%TEI)	0.33 ± 1.14	0.42 ± 1.51	0.26 ± 0.94	0.565
**Grouped dietary data**				
**Dietary data: Morning**				
Morn MEI (%TEI)	23.08 ± 7.45	22.59 ± 7.06	24.04 ± 9.04	0.465
Morn CH (%TEI)	11.98 ± 4.19	11.87 ± 4.02	11.59 ± 4.80	0.884
Morn Pr (%TEI)	3.12 ± 1.44	3.06 ± 1.22	3.43 ± 1.58	0.207
Morn Lip (%TEI)	7.20 ± 3.36	6.86 ± 3.42	8.07 ± 3.69	0.082
**Dietary data: Evening**				
Eve MEI (%TEI)	37.15 ± 9.18	37.33 ± 10.26	36.62 ± 9.86	0.901
Eve CH (%TEI)	12.68 ± 4.78	13.07 ± 4.58	12.70 ± 4.97	0.678
Eve Pr (%TEI)	6.56 ± 2.07	6.59 ± 2.25	6.30 ± 2.51	0.552
Eve Lip (%TEI)	16.21 ± 5.34	15.85 ± 5.80	15.66 ± 5.09	0.742

ADS: After Dinner Snack; B: Breakfast; CC: Common Homozygous; CH: Carbohydrates; CT: Heterozygous; D: Dinner; ES: Evening Snack; Eve: Evening; kJ: Kilojoules; L: Lunch; Lip: Lipids; MEI: Meal Energy Intake; Morn: Morning; MS: Morning Snack; Pr: Proteins; SD: Standard Deviation; TEI: Total Energy Intake; TT: Variant Homozygous; X: Mean.

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
