# Peer review of "Polymorphism of CLOCK Gene rs3749474 as a Modulator of the Circadian Evening Carbohydrate Intake Impact on Nutritional Status in an Adult Sample"

_nutrients, 2020, doi:10.3390/nu12041142_

Round 1

Reviewer 1 Report

In this paper, evaluated the distribution of energy intake and macronutrients consumption throughout the day and how its effect on nutritional status can be modulated by the presence of the rs3749474 polymorphism of the CLOCK gene in the Cantoblanco Platform for Nutritional Genomics.

  1. Is the dependent variable (BMI) in the linear models normally distributed?
  2. Authors should explain why they collected a lot of personal, social and health data but only put two variables (age and sex) in the models. 

Author Response

We would like to thank you for your helpful comments with regard to submission of our paper entitled " Polymorphism of CLOCK gene rs3749474 as a modulator of the circadian evening carbohydrate intake impact on nutritional status in an adult sample." (Manuscript ID nutrients-775055):

Our comments are in the attached document Response Reviewer 1

Changes are included in the template using the "Track Changes" function.

Reviewer 2 Report

I commend you on an excellent and intersting research paper on the implications of some specific genetic influences on weight and health- and draw your attention to some minor (but important) issues which should be addressed.

Minors

Line 35 abtstract take out the word 'the' its not needed here.

Line 95- perhaps 'basic illnesses' is not the best description- minor illnesses?

Line 100- which is it Omron in the Uk or Kyoto Japan? it can't be both...

Line 153 - change 'was' for 'were'

Line 242 exchange investigation for 'research' 

line 245-250- you split the data by genotype grouping (which is essenital of course) but here you are reporting differentiation in lipid profile and glycaemic status by BMI grouping- please report this in the results/show the significance level of the difference- ignore this your inclusion in the appendix will be fine.

Line 314-317 really doesn't make sense and definitely needs re-wording- I think you are saying: the sample uses self-report data and therefore this data is subject to under/over-reporting - so to mitigate against this you included people who's weight stability matched their reported energy intake which you describe as 'plausible reporters'.

Author Response

We would like to thank you for your helpful comments with regard to submission of our paper entitled " Polymorphism of CLOCK gene rs3749474 as a modulator of the circadian evening carbohydrate intake impact on nutritional status in an adult sample." (Manuscript ID nutrients-775055):

Our comments are in the attached document Response Reviewer 2

Changes are included in the template using the "Track Changes" function.
